# End-to-End Differentiable Physics
# for Learning and Control

**Filipe de A. Belbute-Peres**
School of Computer Science
Carnegie Mellon University
Pittsburgh, PA 15213
filiped@cs.cmu.edu

**Kevin A. Smith**
Brain and Cognitive Sciences
Massachusetts Institute of Technology
Cambridge, MA 02139
k2smith@mit.edu

**Kelsey R. Allen**
Brain and Cognitive Sciences
Massachusetts Institute of Technology
Cambridge, MA 02139
krallen@mit.edu

**Joshua B. Tenenbaum**
Brain and Cognitive Sciences
Massachusetts Institute of Technology
Cambridge, MA 02139
jbt@mit.edu

**J. Zico Kolter**
School of Computer Science
Carnegie Mellon University and
Bosch Center for Artificial Intelligence
Pittsburgh, PA 15213
zkolter@cs.cmu.edu

## Abstract

We present a differentiable physics engine that can be integrated as a module in deep neural networks for end-to-end learning. As a result, structured physics knowledge can be embedded into larger systems, allowing them, for example, to match observations by performing precise simulations, while achieves high sample efficiency. Specifically, in this paper we demonstrate how to perform backpropagation analytically through a physical simulator defined via a linear complementarity problem. Unlike traditional finite difference methods, such gradients can be computed analytically, which allows for greater flexibility of the engine. Through experiments in diverse domains, we highlight the system's ability to learn physical parameters from data, efficiently match and simulate observed visual behavior, and readily enable control via gradient-based planning methods. Code for the engine and experiments is included with the paper.[1]

## 1 Introduction

Physical simulation environments, such as MuJoCo [Todorov et al., 2012], Bullet [Coumans et al., 2013], and others, have played a fundamental role in developing intelligent reinforcement learning agents. Such environments are widely used, both as benchmark tasks for RL agents [Brockman et al., 2016], and as "cheap" simulation environments that can (ideally) allow for transfer to real domains. However, despite their ubiquity, these simulation environments are in some sense poorly suited for deep learning settings: the environments are not natively *differentiable*, and so gradients (e.g., policy gradients for control tasks, physical property gradients for modeling fitting, or dynamics Jacobians

for model-based control) must all be evaluated via finite differencing, with some attendant issues of speed and numerical stability. Recent work has also proposed the development of a differentiable physical simulator [Degrave et al., 2016], but this was accomplished by simply writing the simulation engine entirely in an automatic differentiation framework; the limitations of this framework meant that the system only supported balls as objects, with limited extensibility.

In this paper, we propose and present a differentiable two-dimensional physics simulator that addresses the main limitations of past work. Specifically, like many past simulation engines, our system simulates rigid body dynamics via a linear complementarity problem (LCP) [Cottle, 2008, Cline, 2002], which computes the equations of motion subject to contact and friction constraints. In addition to this, however, in this paper we also show how to differentiate, analytically, through the *optimal solution* to the LCP; this allows us to use general simulation methods for determining the non-differentiable parts of the dynamics (namely, the presence or absence of collisions between convex shapes), while still providing a simulation environment that is end-to-end differentiable (given the observed set of collisions). The end result is that we can embed an entire physical simulation environment as a "layer" in a deep network, enabling agents to both learn the parameters of the environments to match observed behavior *and* improve control performance via traditional gradient-based learning. We highlight the utility of this system in a wide variety of different domains, each highlighting a different benefit that such differentiable physics can bring to deep learning systems: learning physical parameters from data; simulating observed (visual) behavior with minimal data requirements; and learning physical deep RL tasks, ranging from pure physical systems like Cartpole to "physics based" like Atari *Breakout*, via gradient planning methods. The environment itself is implemented as a function within popular the PyTorch library [Paszke et al., 2017]. Code for the engine and experiments is available at `https://github.com/locuslab/lcp-physics`.

## 2 Background and related work

The work in this paper relates in some way to several different threads of recent work in deep learning and cognition.

**Physical simulation**  As mentioned above, although they were not developed purely within the machine learning community, physical simulation tools such as MuJoCo [Todorov et al., 2012], Bullet [Coumans et al., 2013], and DART [Lee et al., 2018], have become ubiquitous tools for the creation and development of deep RL agents. In general, these environments also use LCP techniques to compute equations of motion, though often with additional enhancements such as a $O(n)$-time simulation for open chains via Featherstone's algorithm [Featherstone, 1984]. Despite their power, computing derivatives through these engines mostly involves using finite differences, that is, evaluating the forward simulation multiple times with small perturbations to the relevant parameters to approximate the relevant gradients. This strategy is often impractical due to (1) the high computational burden of finite differencing when computing the gradient with respect to a large number of model/policy parameters; and (2) the instability of numerical gradients over long time horizons, especially if contacts change over the course of a rollout. The analytic LCP differentiation approach has no such issues, and can give gradients with respect to a large number of parameters essentially "for free" given a forward solution. The usage of analytical gradients in physics simulation has been previously investigated in spring-damper systems [Hermans et al., 2014]. However, due to its limitations, such as instability and unrealistic contact handling, most engines used in practice do not use spring-damper models.

In probably the most closely related work, Degrave et al. [2016] also develop a differentiable physics engine, with motivations similar to our own. However, in this case they made their engine differentiable by simply implementing it in its entirety in the Theano framework [Al-Rfou et al., 2016]. This severely limited the complexity of the allowable operations: for instance the engine only allowed for collision between balls and the ground plane. In contrast, because our method analytically differentiates the LCP, it can be substituted within the traditional computations of most existing physics engines, only requiring the added differentiability within the LCP portion itself; indeed, in our system we use existing efficient methods for portions of the simulator such as a collision detection or Euler stabilization. Finally, we also evaluate the method in a broader context than in this previous work: while the approach there centered around policy optimization (within the physics engine itself),

we additionally highlight applications in system identification, prediction in visual settings, and using the simulation engine internally *within* a policy to perform tasks in a *different* environment.

**Intuitive physics**  In a related but orthogonal body of work, many studies have investigated the human ability to intuitively understand physics. Battaglia et al. [2013], Hamrick et al. [2015] and Smith and Vul [2013] suggested that people have an "intuitive physics engine" that they can use to simulate future or hypothetical states of the world for inference and planning. Recent work in machine learning has leveraged this idea by attempting to design networks that can learn physical dynamics in a differentiable system [Lerer et al., 2016, Chang et al., 2016, Battaglia et al., 2016], but because these dynamics must be learned, they require extensive training before they can be used as a layer in a larger network, and it is not clear how well they generalize across tasks. Conversely, by performing explicit simulation (similar to how people do), which is embedded as a "layer" in the system, our approach requires no pre-training and can generalize across scenarios that can be captured by a rigid-body engine.

**Model-based reinforcement learning**  Although focusing on an orthogonal issue, our work is of course highly relevant to the entire field of model-based RL. Although model-free RL methods have achieved some notable successes [Mnih et al., 2015, Heess et al., 2015], model-based RL also underpins much of the recent success, and there is both old [Atkeson and Santamaria, 1997] and recent [Kurutach et al., 2018] work that argues that model-based approaches are often superior for many tasks.

Model-based (deep) RL typically focuses on one of two settings: either a general neural network is used to simulate the dynamics (e.g. Werbos [1989], Nagabandi et al. [2017]), often with a specific loss function or network structured aimed at predicting on the relevant time scales [Abbeel and Ng, 2005, Mishra et al., 2017], or the model used is a more "pure" physics model, where learning essentially corresponds to traditional system identification [Ljung, 1999]. Our approach lies somewhere in between these two extremes (though closer to the system identification approach), where we can use the differentiability of the simulation system to identify model parameters *and* use the system within a model-based control method, but where the generic formulation is substantially more general than traditional system identification, and e.g. the number of objects or joints can even be dictated by another portion of the network.

**Analytic differentiation of optimization**  Finally, our work relates methodologically to a recent trend of incorporating more structured layers within deep networks. Specifically, recent work has looked incorporating quadratic programs [Amos and Kolter, 2017], combinatorial optimization [Djolonga and Krause, 2017], computing equilibria in games [Ling et al., 2018], or dynamic programming [Mensch and Blondel, 2018]. Our work relates most closely to that of [Amos and Kolter, 2017]. Like this work, we use an interior point primal dual method to solve a nonlinear set of equations (in our case a general LCP, in their case a symmetric LCP resulting from the KKT conditions of QP). However, both the general nature of the LCP, and the application to physical simulation, specializes substantially from what has been considered in previous work.

## 3  Differentiable Physics Engine

A detailed description of the physics engine architecture is presented in Appendix A due to space constraints. The LCP solution and the gradients are presented in more detail in Appendix B. Below we present a brief summary of the LCP formulation.

### 3.1  Formulating the LCP

Rigid body dynamics are commonly formulated as a linear complementarity problem, with the different constraints on the movement of bodies (such as joints, interpenetrations, friction, etc.) represented as equality and inequality constraints [Anitescu and Potra, 1997, Cline, 2002]. In this work, we follow closely the framework described in Cline [2002], in which at each time step an LCP is solved to find the constrained velocities of the objects.

To formulate such an LCP, we first find which contacts between bodies are present at the current time-step. Let $t$ be the current time-step and $t + dt$ the following time-step, for a step of size $dt$. If

the distance between possibly contacting objects is less than a predefined threshold, the interaction is considered a contact. From the equality constraints specified in the system, such as joints, we can build the matrix $\mathcal{J}_e$ such that $\mathcal{J}_e v_{t+dt} = 0$. From the contacts at each step, we can build a contact constraint matrix $\mathcal{J}_c$, such that $\mathcal{J}_c v_{t+dt} \geq 0$. Similarly, we have a friction constraint matrix $\mathcal{J}_f$ that introduces frictional interactions. From the definition of the simulated bodies we construct the inertia matrix $\mathcal{M}$. The structure of these block matrices is described in detail in Appendix A. Finally, given the forces acting on the bodies at time $t$, $f_t$, and the collision coefficient $c$, the constrained dynamics can be formulated as the following mixed LCP

$$
\begin{bmatrix} 0 \\ 0 \\ a \\ \sigma \\ \zeta \end{bmatrix} - \begin{bmatrix} \mathcal{M} & -\mathcal{J}_e & -\mathcal{J}_c & -\mathcal{J}_f & 0 \\ \mathcal{J}_e & 0 & 0 & 0 & 0 \\ \mathcal{J}_c & 0 & 0 & 0 & 0 \\ \mathcal{J}_f & 0 & 0 & 0 & E \\ 0 & 0 & \mu & -E^T & 0 \end{bmatrix} \begin{bmatrix} v_{t+dt} \\ \lambda_e \\ \lambda_c \\ \lambda_f \\ \gamma \end{bmatrix} = \begin{bmatrix} \mathcal{M}v_t + dtf_t \\ 0 \\ c \\ 0 \\ 0 \end{bmatrix} \tag{1}
$$

$$
\text{subject to } \begin{bmatrix} a \\ \sigma \\ \zeta \end{bmatrix} \geq 0, \begin{bmatrix} \lambda_c \\ \lambda_f \\ \gamma \end{bmatrix} \geq 0, \begin{bmatrix} a \\ \sigma \\ \zeta \end{bmatrix}^T \begin{bmatrix} \lambda_c \\ \lambda_f \\ \gamma \end{bmatrix} = 0,
$$

where $[a, \sigma, \zeta]^T$ are slack variables for the inequality constraints, and $[v_{t+dt}, \lambda_e, \lambda_c, \lambda_f, \gamma]^T$ are the unknowns. By solving this LCP, we obtain the velocities for the next time-step $v_{t+dt}$, which are used to update the positions of the bodies.

## 3.2 Solving the LCP

Analogously to the differentiable optimizer in OptNet [Amos and Kolter, 2017], our LCP solver is adapted from the primal-dual interior point method described in Mattingley and Boyd [2012]. The advantage of using such a method is that it allows for efficient computation of the gradients, as we show in Section 3.3.

First, to simplify the notation from the LCP formulation of the dynamics in Equation 1, let us define

$$
\begin{aligned}
&x := -v_{t+dt} && q := -\mathcal{M}v_t - dtf_t && s := \begin{bmatrix} a \\ \sigma \\ \zeta \end{bmatrix} && \\
&y := \lambda_e && A := \mathcal{J}_e && && F := \begin{bmatrix} 0 & 0 & 0 \\ 0 & 0 & E \\ \mu & -E^T & 0 \end{bmatrix}. \\
&z := \begin{bmatrix} \lambda_c \\ \lambda_f \\ \gamma \end{bmatrix} && G := \begin{bmatrix} \mathcal{J}_c & 0 \\ \mathcal{J}_f & 0 \\ 0 & 0 \end{bmatrix} && m := \begin{bmatrix} c \\ 0 \\ 0 \end{bmatrix} &&
\end{aligned}
$$

Then we can rewrite the LCP above as the system below, which can be solved with only slight adaptations to the primal-dual interior point method by [Mattingley and Boyd, 2012].

$$
\begin{bmatrix} 0 \\ s \\ 0 \end{bmatrix} + \begin{bmatrix} \mathcal{M} & G^T & A^T \\ G & F & 0 \\ A & 0 & 0 \end{bmatrix} \begin{bmatrix} x \\ z \\ y \end{bmatrix} = \begin{bmatrix} -q \\ m \\ 0 \end{bmatrix} \tag{2}
$$

$$
\text{subject to } s \geq \mathbf{0}, \ z \geq \mathbf{0}, \ s^T z = 0.
$$

## 3.3 Gradients

All the work leading to the construction of the dynamics LCP in Equation 1 consists of differentiable operations on the simulations parameters and initial setting. Therefore, if we could differentiate through the solution for the LCP as well, the system would be differentiable end to end. To derive these gradients we apply the method described in [Amos and Kolter, 2017] to the LCP in 2, which gives us the gradients of the solution of the LCP with respect to the input parameters from the previous time-step. By following this method we arrive at the partials that can then be used for the backward

step

$$\frac{\partial \ell}{\partial q} = -d_x \qquad\qquad \frac{\partial \ell}{\partial \mathcal{M}} = -\frac{1}{2}(d_x x^T + x d_x^T)$$

$$\frac{\partial \ell}{\partial m} = D(z^\star) d_z \qquad\qquad \frac{\partial \ell}{\partial G} = -D(z^\star)(d_z x^T + z d_x^T) \qquad (3)$$

$$\frac{\partial \ell}{\partial A} = -d_y x^T - y d_x^T \qquad\qquad \frac{\partial \ell}{\partial F} = -D(z^\star) d_z z^T.$$

### 3.4 Implementation

The physics engine is implemented in PyTorch [Paszke et al., 2017] in order to take advantage of the autograd automatic differentiation graph functionality. The LCP solver is implemented as an autograd Function, with the analytical gradients provided according to the definitions above. This allows the derivatives to be propagated across time-steps in the simulation. Furthermore, the autograd graph then allows the derivatives to be propagated backwards into the leaf parameters of the dynamics, such as the bodies' masses, positions, etc.

## 4 Experiments

To demonstrate the flexibility of the differentiable physics engine, we test its performance across three classes of experiments. First, we show that it can infer the mass of an object by observing the dynamics of a scene. Next, we demonstrate that embedding a differentiable physics engine within a deep autoencoder network can lead to high accuracy predictions and improved sample efficiency. Finally, we use the differentiable physics engine together with gradient-based control methods to show that we can learn to perform physics-based tasks with low sample complexity when compared to model-free methods.

### 4.1 Parameter learning

**Task**   To evaluate the engine's capabilities for inference, we devised an experiment where one object has unknown mass which has to be inferred from its interactions with the other bodies. As depicted in Figure 1, a scene in which a ball of known mass hits a chain is observed and the resulting positions of the objects are recorded for $10s$. The goal is to infer the mass of the chain.

**Learning and results**   Simulations are iteratively unrolled starting with an arbitrarily chosen mass of 1 for the chain. After each iteration, the mean squared error (MSE) between the observed positions and the simulated positions is observed, and then used to obtain its gradient with respect to the mass. Gradients are clipped to a maximum absolute value of 100 and then used to perform gradient descent on the value of the mass, with a learning rate of 0.01. As shown in Figure 1 this process is able to quickly reduce the position MSE by converging to the true value of the mass.

**Comparison to numerical derivatives**   We also compared using analytic and numerical gradients. In this experiment, the same optimization process described above was repeated for a varying number of links in the chain. The number of gradient updates was fixed to 30 and the run times were averaged over 5 runs for each condition. As can be seen in Figure 1, the run time with analytical gradients grows much more slowly with the increasing number of parameters.

### 4.2 Prediction on visual data

**Task**   To test our approach on a benchmark for visual physical prediction, we generated a dataset of videos of billiard ball-like scenarios using the code from [Fragkiadaki et al., 2015]. Simulations lasting 10 seconds were generated, totalling 8,000 trials for training, 1,000 for validation and 1,000 for testing. Datasets with 1 and 3 balls were used, with all balls having the same mass. Frames from sample trials can be seen in Figure 3. In our task setup, balls bouncing in a box are observed for a period of time. The model is provided with 3 frames as input and has to learn to predict the state of the world at a future state, 10 frames later.

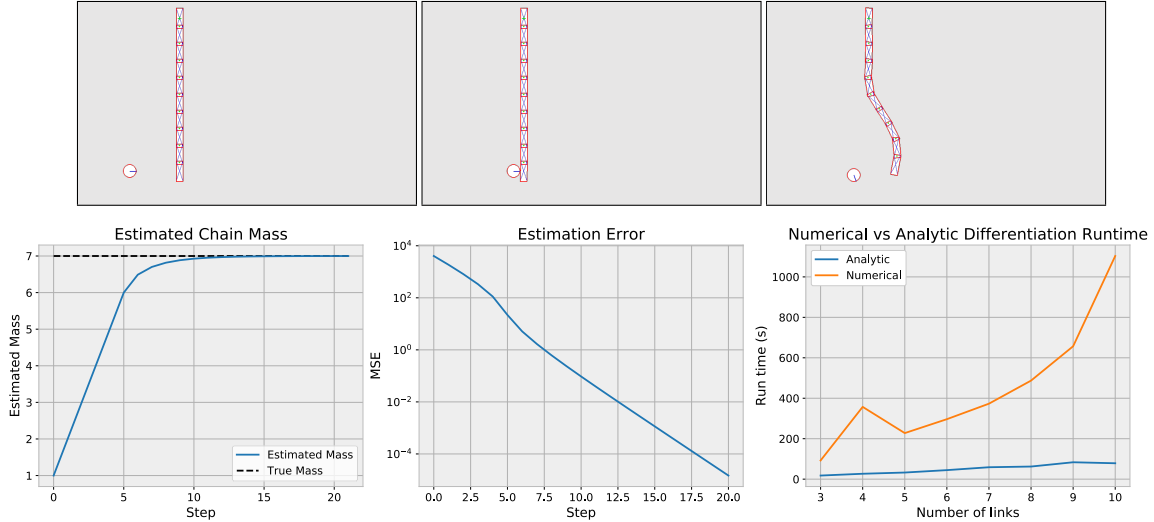

Figure 1: Inferring the mass of a chain. *Top*: Sequence of frames from the inference experiment. The goal is to infer the mass of the chain by unrolling simulations and using the gradient to minimize the loss from the predicted positions to the observed ones. *Bottom left*: The estimated mass quickly converges to the true value, $m = 7$, indicated by the dashed line. *Bottom center*: As a consequence of the improving mass estimation, the MSE (represented in log scale) between the true and simulated positions for the bodies decreases quickly. *Bottom right*: Run time comparison between using analytical gradients or finite differences for 30 updates, as a function of the number of links in the chain.

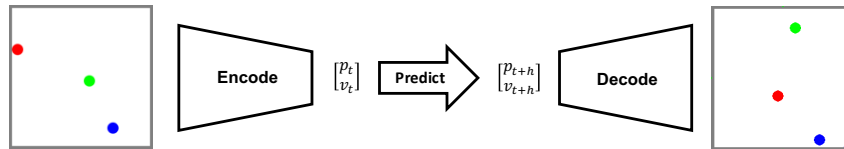

Figure 2: Diagram of autoencoder architecture. The encoder learns to map from input frames to the physical state of the objects (*i.e.*, position, velocity, etc.). The physics engine steps the world forward using the parameters from the encoder. The decoder takes the predicted physical parameters and generates a frame to match the true future frame. The system is trained end-to-end. Part of the labels have strong supervision, with ground truth values available for the output of the encoder and physics engine. Different proportions of strong and weak supervision (only the future frame is provided) in the data are evaluated. Using a large number of weakly labelled data improves sample efficiency for strongly labelled data.

**Architecture**    To make visual prediction given the visual input, we use an autoencoder architecture summarized in Figure 2. It consists of three parts: (1) the *encoder* maps input frames to the physical state of the objects (*i.e.*, position, velocity, etc.). Specifically, we take in a sequence of 3 RGB frames from the simulation. We then use a pretrained spatial pyramid network [Ranjan and Black, 2016] to obtain two optical flow frames (each consisting of two matrices, for x and y flow). Color filters are applied to the RGB images to segment the objects. The segmented region of each object is then used as a mask for the RGB and optical flow frames, such that at the end of this pipeline we have, for each object, a collection of 3 RGB frames and 2 optical flow frames (13 channels) with only a single segmented object. Then, each of these per-object processed inputs is passed to a VGG-11 network with its last layer modified to output size 4, in order to regress two position and two velocity parameters as outputs. (2) the *physics engine* steps the world forward using the physical parameters received from the encoder. The physics engine can be integrated into the autoencoder pipeline and allow for end-to-end training due to its differentiability, as described in Section 3. (3) the *decoder* takes the predicted physical parameters and generates a frame to match the true future frame. The architecture used is a mirror of the VGG encoder network, with transposed convolutions in the place of convolutions and bilinear upsampling layers in the place of the maxpooling ones.

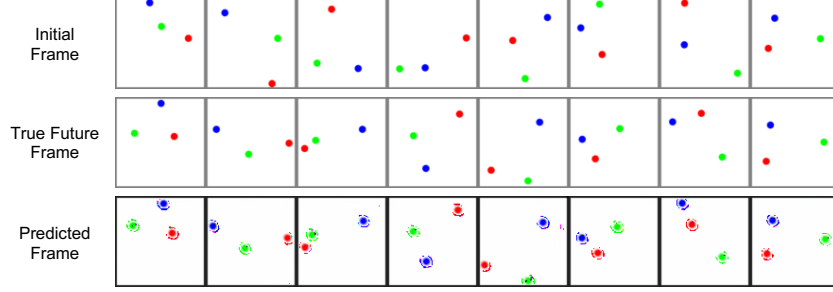

Figure 3: Qualitative results for prediction task comparing ground truth and predicted future frame. Only the initial frame and the two preceding frames are used as input, with physical parameters extracted, used to simulated the state forward and generate the predicted frame. In most cases the predicted frame is accurate. However, small differences can still be perceived in some cases, due to differences between the two engines.

**Learning** In order to evaluate the sample efficiency of the model, the network was trained with varying amounts of labelled samples. The labels used consist of the ground truth physical parameters $\phi$ of the objects both at the present ($\phi_t$) and the future time-step ($\phi_{t+dt}$). When a label is available for a given sample, the model uses these ground truth physical parameters (instead of the estimated ones) to generate the predicted frame $\hat{y}$ from input frames $x$, such that

$$\hat{\phi}_t = encoder(x), \quad \hat{\phi}_{t+dt} = physics(\phi_t), \quad \hat{y} = decoder(\phi_{t+dt}). \qquad (4)$$

Using the labels and the true future frame $y$, the model is then trained to minimize a loss consisting of the sum of three terms, the encoder, physics and decoder losses

$$\mathcal{L} = \mathcal{L}_{enc} + \mathcal{L}_{phys} + \mathcal{L}_{dec},$$
$$\mathcal{L}_{enc} = \ell(\hat{\phi}_t, \phi_t), \quad \mathcal{L}_{phys} = \ell(\hat{\phi}_{t+dt}, \phi_{t+dt}), \quad \mathcal{L}_{dec} = \ell(\hat{y}, y), \qquad (5)$$

where $\ell(\cdot, \cdot)$ is the mean squared error loss.

When labels are not available for a given sample, the model uses its own estimated parameters to generate the predicted frame, that is

$$\hat{\phi}_t = encoder(x), \quad \hat{\phi}_{t+dt} = physics(\hat{\phi}_t), \quad \hat{y} = decoder(\hat{\phi}_{t+dt}). \qquad (6)$$

Notice that here, unlike in Equation 5, the arguments to the function are estimated ($\hat{\phi}_t$, $\hat{\phi}_{t+dt}$). In this case, since there are no labels to use for the other losses, the loss consists only of $\mathcal{L} = \mathcal{L}_{dec}$. Notice that here the right hand side of the equations use the estimated $\hat{\phi}$. The gradients are thus being propagated end-to-end through the physics model back to the encoder. As shown in Figure 4, this signal from unlabeled examples allows the autoencoder to learn with greater sample efficiency. For all losses, the MSE is used. In our experiments, the squared loss performed better than the $\ell_1$ loss, which was not able to produce meaningful decoder outputs.

**Results** As demonstrated in Figure 4, the model was able to learn to perform the task with high accuracy. Figure 3 contains sample predicted frames and their matching ground truth frame for a qualitative analysis of the results. As a comparison point, an MLP with two hidden-layers of size 100 and trained with only labeled data was used as a baseline, replacing the $physics(\cdot)$ function in Equation 4 above. In our experiments, using the baseline model in such a way, as a replacement for the physics function, provided better results than using it in an unstructured manner, relying solely on the decoder loss. It is clear from Figure 4 that the autoencoder with the physics model is able to learn more efficiently and with higher accuracy than the baseline model. To evaluate the sample efficiency of this model, we compare its performance on training regimens in which 100%, 25%, 10% and 2% of the available samples containing labels. Some supervision is still necessary, since when provided with no supervision at all (a 0% condition), relying solely on the decoder loss, the model was not able to learn to extract meaningful physical parameters. Still, as can be seen in Figure 4, the model is able to leverage the unlabeled data to quickly learn even from few labelled data points.

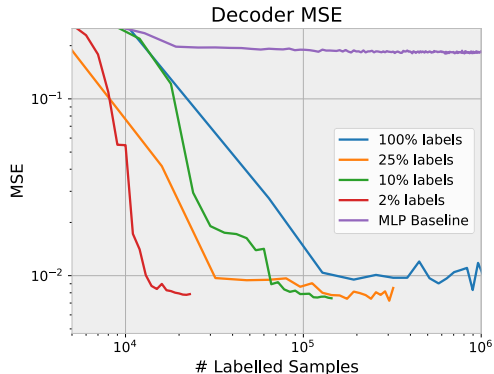

Figure 4: Sample efficiency for the prediction task measured by the validation loss per number of labelled samples used in training. The autoencoder is able to leverage unlabelled examples to improve its sample efficiency: training regimes that employ unlabeled data learn faster for a given amount of labeled data. The loss is the mean squared error of the predicted image to the ground truth. Each line represents a training regiment with a different proportion of labeled to unlabeled data. Note that the x-axis is already adjusted to the number of labeled samples used, to facilitate comparisons.

## 4.3 Control

**Tasks**  Finally, in this section we demonstrate the physics engine ability to be readily used with gradient-based control methods. To this end, we evaluate its performance on physics based tasks from the OpenAI Gym's environment [Brockman et al., 2016], namely *Cartpole* and the Atari game *Breakout*.

**Model and Controller**  For the *Cartpole* environment, a model is built using two articulated rectangles, whose dimensions and mass are learned from simulated trajectories using random actions. The physics engine-based model is compared to a baseline consisting of an MLP with two hidden layers of size 100 trained on the same data. A variation of the environment is used in which the actions to be taken by the cart are continuous, instead of discrete. Rewards are also limited to 1000, instead of the default 200 for which the task is considered done.

For *Breakout*, a model of the environments is built by applying color filters, segmenting the diverse objects (the paddle, the ball, etc.) and translating these positions into the physics engine. The ball's velocity is estimated by the difference in its position from the last two frames. The paddle velocity when moving at each step is learned by unrolling game episodes with randomly chosen actions, performing the same actions in the physics simulation and then fitting the simulation parameter via gradient descent to minimize the mean squared error to the observed trajectory, analogously to the process in Section 4.1. The physics engine model is compared to a Double Q Learning with prioritized replay [van Hasselt et al., 2015] baseline from OpenAI [Dhariwal et al., 2017].

Since the resulting physics models described above are differentiable, they are used in conjunction with iLQR [Li and Todorov, 2004] to control the agent in the tasks. The iLQR is set up with a time-horizon of 5 frames for both tasks. For *Cartpole* the cost consists of the square of pole's angular deviation from vertical. For the Ataro game the cost consists of the squared difference in the x position of the paddle and the ball when the ball is descending, and the squared distance to the center of the screen otherwise.

**Results**  Results for the *Cartpole* task are shown in Figure 5. Even though the MLP baseline achieves a lower MSE faster in predicting the next state of the cartpole system, the physics engine is able to learn parameters for a model that allows for high reward on the task, even when error is higher.

In the Atari benchmark, the system is able to achieve high reward on the task with extremely low sample complexity. Specifically, the model is able to learn the paddle parameters quickly from random trajectories, improving the control precision, and leading to high reward, as shown in Figure 6

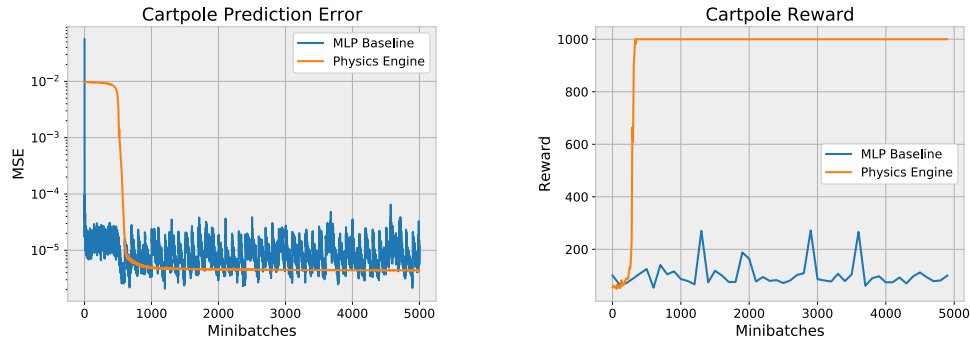

Figure 5: Even though the baseline is able to achieve lower MSE over one-step predictions of the dynamics of the *Cartpole* environment (*left*), the physics engine-based controller is able to achieve a higher reward very quickly (*right*).

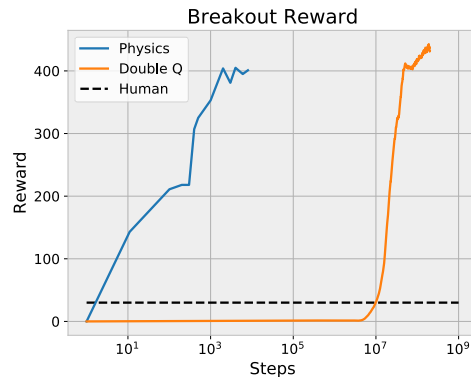

Figure 6: The physics based controller is able to quickly learn a good parameter values that lead to high reward. Even though the asymptotic performance is lower than the model-free method, it achieves a high level of reward with orders of magnitude data (the horizontal axis is log-scaled). Human level of 31 for a professional game tester was used, as per [Mnih et al., 2015].

for *Breakout*. The model performs close to model-free reinforcement learning methods and is able to achieve a high level of reward with orders of magnitude fewer samples.

# 5   Conclusion

In this work, we have presented a differentiable physics engine. Unlike most previous work, this engine provides analytical gradients by differentiating the solution to the physics LCP. The differentiable nature of the engine allows it to be used as a part of gradient-based learning and control system systems. Our experiments demonstrate the diverse possibilities this system entails, such as inferring parameters from observed simulations, learning from visual observations and performing accurate predictions, and achieving high reward with gradient-based control methods on physics-based tasks, all the while demonstrating sample efficiency. This modular, differentiable system contributes to a recent trend of incorporating components with structured modules into end-to-end learning systems, such as deep networks. In general, we believe structured constraints such as the ones provided by physics simulation are essential for providing a scaffolding for more efficient learning.

## Footnotes

[1]Available at https://github.com/locuslab/lcp-physics.

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
