[Supplementary Material · Supplementary.pdf]

# A Physics Engine

In this section, we present a description of the structure of the physics engine. The formulation of the physics engine described here follows closely the one presented by Cline [2002], with some simplifications applied due to the engine presented here being two-dimensional. The description presented here is brief and intended only to be a sufficient guide to reproducing the work in the paper. For a more detailed introduction to physics engines, including comparisons to other architectural choices, see Garstenauer and Kurka [2006].

## A.1 Step Overview

A physics simulation proceeds over time by iteratively taking small steps of size $dt$. In this section, we describe conceptually the sequence of sub-steps that compose a step in the simulation. In the following sections, we describe in greater depth each of these parts.

1. At the beginning of time step $t$ we have the bodies at positions $p_t$ with velocities $v_t$, as defined in Section A.3 (Equation 9). Importantly, at this point (the beginning of the step), we assume current contacts are known and that constraints are satisfied (*i.e.*, there are no interpenetrations).

2. External forces acting on bodies are added up to form the force vector $f_t$, as defined in Section A.3 (Equation 10).

3. Constraint matrices for the current step are formed, as defined in Section A.4.

4. We solve the dynamics LCP defined in Section A.5 to get the velocities $v_{t+dt}$.

5. Numerical integration is used with the velocities $v_{t+dt}$ to get the positions $p_{t+dt}$. For example, simple explicit Euler integration gives $p_{t+dt} = p_t + dt \cdot v_{t+dt}$. The new positions $p_{t+dt}$ have contacts detected and checked for interpenetrations (details in Section A.6). If interpenetrations do occur, we divide $dt$ in half and repeat the numerical integration from $p_t$ to $p_{t+dt}$ with the new $dt$. We repeat this process until a position free of interpenetrations is found. Since we know there are no interpenetrations at the beginning of the step (by the assumption in sub-step 1), we know there is $dt > 0$ for which there are no interpenetrations.

6. If post-stabilization is being employed (described in Section A.7), then constraint matrices are re-calculated from the new contacts at positions $p_{t+dt}$, and the post-stabilization correction to the positions is applied. Importantly, the post-stabilization update does not violate constraints, thus we still end the step with no interpenetrations.

## A.2 Bodies

The basic unit in the engine are the rigid bodies. Bodies possess mass, position and velocity, and forces act on them. The position and velocity of a body are composed of three components: an angular ($a$) components, and two linear ($x$ and $y$) components,

$$p_{body} = \begin{bmatrix} p_a \\ p_x \\ p_y \end{bmatrix} \qquad v_{body} = \begin{bmatrix} v_a \\ v_x \\ v_y \end{bmatrix}. \qquad (7)$$

The mass of a body, $m$, defines the mass-inertia matrix of a body, $\mathcal{M}_{body}$, which is given by

$$\mathcal{M}_{body} = \begin{bmatrix} \mathcal{I} & 0 & 0 \\ 0 & m & 0 \\ 0 & 0 & m \end{bmatrix},$$

where $\mathcal{I}$ is the moment of inertia for the body, which is a scalar in 2D. The moment of inertia is a function of the mass and the shape of a body. For example, for a circle, we have $\mathcal{I} = \frac{1}{2}mr^2$, where $r$ is the radius.[2]

Moreover, bodies also possess dimensionless scalar parameters that define their behavior when in contact with other bodies, namely the collision restitution coefficient and the friction coefficient. The

restitution coefficient $k$ specifies the elasticity of the collision, with $k = 1$ specifying a perfectly elastic collision, and $k = 0$ specifying a perfectly inelastic collision. The friction here is defined by a single coefficient $\mu$, with no distinction for static and dynamic friction. When two bodies are in contact, the frictional force opposes a body's movement of sliding against the other. The friction coefficient coefficient defines the maximal frictional force as a proportion to the normal force between the bodies, that is

$$f_{fric} \leq \mu f_{normal}.$$

Finally, each body has a set of external forces that act on it. External forces are represented as acting on the center of mass of the body, and they are represented as a vector with three components: a torque ($\tau$) component, and two linear ($f_x$ and $f_y$) components

$$f_{external} = \begin{bmatrix} \tau \\ f_x \\ f_y \end{bmatrix}. \tag{8}$$

## A.3 Global Parameters

To simplify the simulation equations specified below, parameters for all bodies are grouped into global structures. Assuming there are $n$ bodies in a simulation, ordered consistently from 1 to $n$, the global position and velocity vectors are given by

$$p = \begin{bmatrix} p_1 \\ \vdots \\ p_n \end{bmatrix} \qquad v = \begin{bmatrix} v_1 \\ \vdots \\ v_n \end{bmatrix}. \tag{9}$$

where $p_i$ and $v_i$ are three dimensional position and velocity vectors (as defined in Equation 7) for each body $i \in \{1, \ldots, n\}$. Forces acting on bodies are also concatenate

Similarly, the mass-inertia matrices for all $n$ bodies are concatenated into a large global matrix $\mathcal{M}$, given by

$$\mathcal{M} = \begin{bmatrix} \mathcal{M}_1 & & 0 \\ & \ddots & \\ 0 & & \mathcal{M}_n \end{bmatrix}.$$

A global external force vector is constructed by summing all external forces acting on each body, and concatenating them into a single vector. We thus have

$$f = \begin{bmatrix} f_1 \\ \vdots \\ f_n \end{bmatrix}, \tag{10}$$

where each $f_i$ is the sum of all external forces acting on body $i$.

## A.4 Constraints

In this section we describe in detail the equations that constrain the dynamics of the bodies. These constraints are divided into three categories: equality, contact and friction constraints.

### A.4.1 Equality constraints

Equality constraints take the form $g(v) = 0$, where $g$ is some function of the velocities. These constraints can be used to implement, for example, joints of many kinds.

In general, for two bodies ($a$ and $b$), equality constraints are defined by two Jacobian matrices ($\mathcal{J}_e^{(body)}$), one for each body, such that

$$\mathcal{J}_e^{(a)} v^{(a)} + \mathcal{J}_e^{(b)} v^{(b)} = 0. \tag{11}$$

A hinge joint, for example, which in two dimensions gives the bodies only one degree of freedom to rotate about their connection point, would be defined by the following two Jacobians

$$\mathcal{J}_e^{(a)} = \begin{bmatrix} -r_{a_y} & 1 & 0 \\ r_{a_x} & 0 & 1 \end{bmatrix} \quad \text{and} \quad \mathcal{J}_e^{(b)} = \begin{bmatrix} r_{b_y} & -1 & 0 \\ -r_{b_x} & 0 & -1 \end{bmatrix},$$

where $r_a$ and $r_b$ are the vectors pointing to the connection point from the center of each body. Substituting these two matrices into Equation 11 we can see that we get an equation that constraints the translation velocities of the two bodies at the connection point to be equal.

Finally, to more easily apply all constraint simultaneously in concert with the global parameters defined above, we define a global constraint Jacobian $\mathcal{J}_e$. For $n$ bodies and $m$ constraints, $\mathcal{J}_e$ is formed as a block matrix such that we have

$$\mathcal{J}_e v = \begin{bmatrix} \mathcal{J}_{11} & \cdots & \mathcal{J}_{1n} \\ \vdots & \ddots & \vdots \\ \mathcal{J}_{m1} & \cdots & \mathcal{J}_{mn} \end{bmatrix} \begin{bmatrix} v_1 \\ \vdots \\ v_n \end{bmatrix} = 0, \tag{12}$$

where in each block-row $i$ of $\mathcal{J}_e$ all blocks $\mathcal{J}_{ij}$ are zero matrices, except for the two blocks $\mathcal{J}_{ia}$ and $\mathcal{J}_{ib}$ corresponding to the two bodies we want to constraint, which are then given by Equation 11.

### A.4.2 Contact constraints

Contact constraints are inequality constraints and thus take the form $g(v) \geq 0$. These constraints enforce that rigid bodies do not interpenetrate.

In general, for two bodies ($a$ and $b$), contact constraints are defined by two Jacobian matrices ($\mathcal{J}_c^{(body)}$), one for each body, such that

$$\mathcal{J}_c^{(a)} v_{t+dt}^{(a)} + \mathcal{J}_c^{(b)} v_{t+dt}^{(b)} + c_{ab} \geq 0, \tag{13}$$

where the term $c_{ab}$ depends on the velocities before the contact and on the combined restitution parameter for $a$ and $b$, $k_{ab}$,

$$c_{ab} = k_{ab} \left[ \mathcal{J}_c^{(a)} v_t^{(a)} + \mathcal{J}_c^{(b)} v_t^{(b)} \right]. \tag{14}$$

The combined restitution parameter is usually defined as a simple function of the restitution parameter of each body, for example $k_{ab} = \frac{1}{2}(k_a + k_b)$.

At each time step, contact constraints as defined in Equation 13 are constructed for each pair of bodies in contact. Using the normal vector $n$ and the contact points $p_a$ and $p_b$ provided by the contact detection algorithms (see Section A.6), the Jacobians are $1 \times 3$ matrices defined as[3]

$$\mathcal{J}_c^{(a)} = \begin{bmatrix} (p_a \times n) & n^T \end{bmatrix} \quad \text{and} \quad \mathcal{J}_c^{(b)} = \begin{bmatrix} (p_b \times n) & n^T \end{bmatrix}, \tag{15}$$

As before, we define a global constraint Jacobian $\mathcal{J}_c$. For $n$ bodies and $m$ constraints, $\mathcal{J}_c$ is formed as a block matrix such that we have

$$\mathcal{J}_c v_{t+dt} = \begin{bmatrix} \mathcal{J}_{11} & \cdots & \mathcal{J}_{1n} \\ \vdots & \ddots & \vdots \\ \mathcal{J}_{m1} & \cdots & \mathcal{J}_{mn} \end{bmatrix} \begin{bmatrix} v_1 \\ \vdots \\ v_n \end{bmatrix} \geq -c, \tag{16}$$

where in each block-row $i$ of $\mathcal{J}_c$ all blocks $\mathcal{J}_{ij}$ are zero matrices, except for the two blocks $\mathcal{J}_{ia}$ and $\mathcal{J}_{ib}$ corresponding to the two bodies in contact, which are then given by Equation 15. The term $c$ is then given by

$$c = \text{diag}(k)\mathcal{J}_c v_t,$$

with $k = [k_1, \ldots, k_m]^T$, where $k_i = k_{ab}$ for the two contacting bodies in contact $i$.

### A.4.3 Friction constraints

Friction constraints are also inequality constraints and thus take the form $g(v) \geq 0$. While contact forces act on the normal direction, friction constraints create forces that act tangentially to the plane of contact of two bodies.

For simplicity, in this section we will for now simply describe the structure of the friction Jacobian $\mathcal{J}_f$, noting its similarity to the contact Jacobian $\mathcal{J}_c$. In Section A.5, we will describe the structure of the inequalities and the complementarity constraints that cause the frictional forces to behave as expected.

As mentioned above, frictional constraints act on contacts, but in the tangential directions instead of the normal direction. In two dimensions there are two tangential directions to a contact, the two orthogonal directions to the contact normal vector. Intuitively, we can imagine that the friction Jacobians will have a structure analogous to the contact Jacobians in Equation 15, with the normal vector substituted by the tangent vectors. Since there are two tangent directions, we will have two constraints for each contact. Let us call $d$ the left orthogonal vector to the normal contact vector $n$, and $p_a$ and $p_b$ the contact points, as provided by the contact detection algorithms (see Section A.6). We then have the friction Jacobians for a contact between bodies $a$ and $b$

$$\mathcal{J}_f^{(a)} = \begin{bmatrix} (p_a \times d) & d^T \\ (p_a \times -d) & -d^T \end{bmatrix} \quad \text{and} \quad \mathcal{J}_f^{(b)} = \begin{bmatrix} (p_b \times d) & d^T \\ (p_b \times -d) & -d^T \end{bmatrix}. \tag{17}$$

As before, we define a global constraint Jacobian $\mathcal{J}_f$. For $n$ bodies and $m$ constraints, $\mathcal{J}_c$ is formed as a block matrix given by

$$\mathcal{J}_f = \begin{bmatrix} \mathcal{J}_{11} & \cdots & \mathcal{J}_{1n} \\ \vdots & \ddots & \vdots \\ \mathcal{J}_{k1} & \cdots & \mathcal{J}_{kn} \end{bmatrix},$$

where in each block-row $i$ of $\mathcal{J}_f$ all blocks $\mathcal{J}_{ij}$ are zero matrices, except for the two blocks $\mathcal{J}_{ia}$ and $\mathcal{J}_{ib}$ corresponding to the two bodies in contact, which are then given by Equation 17.

Two other matrices will be important when dealing with friction constraints, $E$ and $\mu$. We will define them here for later reference. If there are $m$ contacts for a given time step, we have

$$E = \begin{bmatrix} e_1 & \cdots & e_n \end{bmatrix} \quad \text{and} \quad \mu = \begin{bmatrix} \mu_1 & & \\ & \ddots & \\ & & \mu_m \end{bmatrix}.$$

Here, $\mu_i \in \mathbb{R}$ is the combined friction coefficient two bodies involved in contact $i$, which can be defined as the average of each body's friction coefficient, for example. Moreover, $e_i \in \mathbb{R}^{2m}$ is a column vector of zeroes except for the two entries $2i - 1$ and $2i$ (e.g., $e_2 = [0, 0, 1, 1, \ldots, 0]^T$ and $e_m = [0, \ldots, 0, 1, 1]^T$).

## A.5 Dynamics LCP

Let us call $\dot{v}$ the acceleration vector. From Newtonian dynamics, it generally holds that

$$\mathcal{M}\dot{v} = f^{(c)} + f,$$

where $f^{(c)}$ are constraint forces inherent to the dynamics, and $f$ are external forces applied to the bodies. These two are here assumed to comprise the totality of forces acting on bodies. However, formulating the dynamics at the acceleration level can lead to systems with no solutions in the presence of friction [Anitescu and Potra, 1997]. Fortunately, by approximating the acceleration with a discrete step

$$\dot{v}_{t+dt} \approx \frac{v_{t+dt} - v_t}{dt},$$

we can rewrite the dynamics equation as

$$\mathcal{M}(v_{t+dt} - v_t) = dt f_t^{(c)} + dt f_t, \tag{18}$$

to get a velocity-based formulation, which is guaranteed to have a solution even with friction constraints. Since $dt$ is a small time-step, $dt f_t^{(c)}$ can be seen as approximate constraint impulses. We

omit the derivation here (see Garstenauer and Kurka [2006]), but these constraint impulses can be written as

$$dt f^{(c)} = \mathcal{J}\lambda, \tag{19}$$

where $\mathcal{J}$ is a constraint Jacobian such as the ones described in Section A.4, and $\lambda$ is some vector of multipliers. By rearranging the terms in Equation 18 and combining with Equation 19 (broken down into the equality, contact and friction constraint matrices), we get the final dynamics equation

$$\mathcal{M}v_{t+dt} - \mathcal{J}_e\lambda_e - \mathcal{J}_c\lambda_c - \mathcal{J}_f\lambda_f = \mathcal{M}v_t + dtf_t. \tag{20}$$

Moreover, for a realistic rigid body simulation, we know that the impulses $\mathcal{J}_c\lambda_c$ can act to push bodies apart and avoid interpenetrations, but they cannot act to pull bodies together. Hence, we must have $\lambda_c \geq 0$. Additionally, if for each constraint $i$, $(\mathcal{J}_cv)_i + c_i = a_i$ for some $a_i > 0$ strictly greater than zero, then the bodies are moving apart and no separating forces should be applied, *i.e.* $(\lambda_c)_i = 0$. Conversely, if this condition is not satisfied, then a separating force is needed to counteract the penetration velocity, thus $(\lambda_c)_i = 0$. This gives rise to the following complementarity condition,

$$\lambda_c \geq 0, \quad a := \mathcal{J}_cv + c \geq 0 \quad \text{and} \quad a^T\lambda_c = 0. \tag{21}$$

The friction terms have similar complementarity constraints. However, due to the nature of Coulomb friction these constraints involve the contact terms (for example, to assert there is no frictional force when contact normal force is 0). We will omit the derivation here (see Cline [2002]), but these interactions give rise to the following constraints,

$$\zeta := \mu\lambda_c - E^T\lambda_f \geq 0, \quad \sigma := \mathcal{J}_fv + \gamma E \geq 0, \quad \sigma^T\lambda_f = 0 \quad \text{and} \quad \zeta^T\gamma = 0, \tag{22}$$

with $\lambda_f \geq 0$ and $\gamma \geq 0$.

Taking the constraints defined in Section A.4 (Equations 12 and 16), the dynamics equation (Equation 20), and the complementarity conditions formulated above (Equations 21 and 22), the dynamics for a step in the simulation can be summarized as the following LCP

$$\begin{bmatrix} 0 \\ 0 \\ a \\ \sigma \\ \zeta \end{bmatrix} - \begin{bmatrix} \mathcal{M} & -\mathcal{J}_e & -\mathcal{J}_c & -\mathcal{J}_f & 0 \\ \mathcal{J}_e & 0 & 0 & 0 & 0 \\ \mathcal{J}_c & 0 & 0 & 0 & 0 \\ \mathcal{J}_f & 0 & 0 & 0 & E \\ 0 & 0 & \mu & -E^T & 0 \end{bmatrix} \begin{bmatrix} v_{t+dt} \\ \lambda_e \\ \lambda_c \\ \lambda_f \\ \gamma \end{bmatrix} = \begin{bmatrix} \mathcal{M}v_t + dtf_t \\ 0 \\ c \\ 0 \\ 0 \end{bmatrix} \tag{23}$$

$$\text{subject to} \begin{bmatrix} a \\ \sigma \\ \zeta \end{bmatrix} \geq 0, \begin{bmatrix} \lambda_c \\ \lambda_f \\ \gamma \end{bmatrix} \geq 0, \begin{bmatrix} a \\ \sigma \\ \zeta \end{bmatrix}^T \begin{bmatrix} \lambda_c \\ \lambda_f \\ \gamma \end{bmatrix} = 0.$$

Where the inequality constraints are written as equality constraints using the slack variables $[a, \sigma, \zeta]^T$ defined above, and $[v_{t+dt}, \lambda_e, \lambda_c, \lambda_f, \gamma]^T$ are the unknowns. From the solution, we obtain the velocities $v_{t+dt}$ for the next step, which are used to update the positions of the bodies as described in Section A.1 (item 5).

## A.6 Contact Detection

Let us define the distance between two bodies as the minimum length between two points, one in surface of each body. Two bodies are then considered to be in contact if the distance between them is less than some parameter $\epsilon > 0$. In other words, for simulation purposes, two bodies are in contact if they are either interpenetrating (*i.e.*, distance smaller than zero), or "touching" (*i.e.*, distance between 0 and $\epsilon$).

The purpose of detecting contacts is to be able to enforce non-interpenetration constraints. From the definitions of the constraint matrices in Section A.4, we can see that for each contact between two bodies, our formulation requires a normal contact vector and a contact point in each body. The normal vector defines the direction in which the contact force will be applied, while the contact points define the points in which such force will be applied in each body.

The process of detecting contacts is divided into two phases: a "broadphase" that cheaply generates candidate contacts by finding bodies that are in each others vicinity, and a "narrowphase" that analyses candidate contacts carefully to determine if they are truly contacts and to generate the

contact information. A naive broadphase approach is to simply list all possible pairs of objects, followed then by a narrowphase will have to verify every possible contact. More efficient broadphase algorithms exist, but these will not be discussed here for the sake of brevity. Refer to Bergen [2004] for a more detailed exposition.

In the narrowphase, as described above, we want to not only verify that a contact between two bodies is present, but also to generate the required information related to that contact. We will rely on two algorithms for this purpose: the Gilbert–Johnson–Keerthi (GJK) and the Separating Axis Theorem (SAT). For a detailed exposition of these algorithms, refer to Catto [2010] and Gregorius [2013]. Suffice it to say here that the GJK algorithm can provide the closest points between two disjoint convex shapes and that the SAT algorithm can provide the axis of minimum penetration between two interpenetrating convex shapes.

The specifics of how contact detection is handled depends on the shapes involved. For the sake of brevity, we will cover here two examples: (1) circle against circle and (2) circle against convex hull. For a detailed exposition containing more collision types, including convex hulls against convex hulls, please refer to Gregorius [2015].

### A.6.1 Circle against circle

For two circles, checking for contacts is simple. The distance between the two bodies, with positions $p_1$ and $p_2$ and radii $r_1$ and $r_2$, is given by

$$d = \|p_1 - p_2\| - r_1 - r_2.$$

We thus have a contact if $d < \epsilon$ and an interpenetration if $d < 0$. The normal vector is simply given by

$$n = \frac{p_1 - p_2}{\|p_1 - p_2\|}.$$

Finally, the contact points (given in each body's reference frame) are

$$c_1 = -n \cdot r_1 \quad \text{and} \quad c_2 = n \cdot r_2.$$

### A.6.2 Circle against convex hull

In this case, we start by applying the GJK algorithm on the convex hull and the center of the circle. There are two possible cases. First, if the circle's center is outsde the hull, GJK will return the point in the convex hull closest to the circle's center. From this, we have that the distance between the two bodies is

$$d = \|p_c - p_{GJK}\| - r,$$

where $p_c$ and $r$ are the circle's center and radius, and $p_{GJK}$ is the nearest point to $p_c$ in the hull. As before, we have a contact if $d < \epsilon$ and an interpenetration if $d < 0$. The normal is then given by

$$n = \frac{p_c - p_{GJK}}{\|p_c - p_{GJK}\|},$$

and the contact points are

$$c_c = -n \cdot r \quad \text{and} \quad c_h = p_{GJK}.$$

The second case happens when the circle's center is inside the hull. In this case, we know we have an interpenetration, but GJK does not provide us with enough information to generate the contact. We thus employ the SAT algorithm to find the axis of minimum penetration. This is done by running the SAT on the axes defined by the normal to each the face of the hull (*i.e.*, the vector perpendicular to the face that points out of the hull). Once we find the face with the smallest distance to $p_c$, the collision normal $n$ is the normal to that face, normalized to have length 1. In this case we know there is a penetration, thus the distance $d < 0$ between the two bodies is given by the distance from the circle's center to the closest face of the hull minus the radius of the circle, or equivalently

$$d = (p_c^{(h)} - p_{vert}) \cdot n - r,$$

where $(p_c - p_{vert}) \cdot n$ takes the vector from one of the vertices of the closest face in the hull ($p_{vert}$) to the center of the circle in the hull's reference frame ($p_c^{(h)}$), and projects it onto the normal ($n$). Finally, the contact points are given by the closest point to the circle in the hull's face, given in each body's reference frame

$$c_h = p_c^{(h)} - n \cdot (d + r) \quad \text{and} \quad c_c = c_h + p_c - p_c,$$

where $p_h$ is the hull's position.

## A.7 Post-Stabilization

As described previously, we want our equality constraints to satisfy $\mathcal{J}_e v_t = 0$. However, due to numerical errors, after the integration sub-step we might end up with $\mathcal{J}_e v_{t+dt} \neq 0$. The effect of this is that over time errors can compound, generating, for example, noticeable drift in joints. To correct for this, we apply post-stabilization at the end of a step in order to try to correct such errors.

In post-stabilization, we want to find a correction vector $\delta$ such that $\mathcal{J}_e(v_{t+dt} + \delta) = 0$, thus correcting any errors that might have accrued. Let us call $g_e := \mathcal{J}_e v_{t+dt}$. Then we want

$$\mathcal{J}_e(v_{t+dt} + \delta) = g_e + \mathcal{J}_e \delta = 0.$$

We still want our update $\delta$ to maintain inequality constraints, so that we still have no interpenetrations after post-stabilization. Let us name the inequality constraint before the post-stabilization update $g_c := \mathcal{J}_c v_{t+dt} + c$. Then, if we call $g_c^+$ the constraint after the update, we want

$$g_c^+ := \mathcal{J}_c(v_{t+dt} + \delta) + c = g_c + \mathcal{J}_c \delta \geq 0.$$

Finally, for $\delta$ to be a valid update, we also need it to satisfy $\mathcal{M}\delta - \mathcal{J}_e \lambda_e - \mathcal{J}_c \lambda_c = 0$, for some $\lambda_e$ and $\lambda_c \geq 0$, so that no forces are introduced into the system. From this we have the LCP

$$\begin{bmatrix} 0 \\ 0 \\ g_c^+ \end{bmatrix} - \begin{bmatrix} \mathcal{M} & -\mathcal{J}_e & -\mathcal{J}_c \\ \mathcal{J}_e & 0 & 0 \\ \mathcal{J}_c & 0 & 0 \end{bmatrix} \begin{bmatrix} \delta \\ \lambda_e \\ \lambda_c \end{bmatrix} = \begin{bmatrix} 0 \\ g_e \\ g_c \end{bmatrix}$$
$$\text{subject to } g_c^+ \geq 0, \lambda_c \geq 0, \lambda_c^T g_c^+ = 0.$$

After solving for $[\delta, \lambda_e, \lambda_c]^T$, we apply the post-stabilization update $p_{t+dt}^+ = p_{t+dt} + dt \cdot \delta$ to get the corrected position vector that reduces the constraint error.

# B  Solution and Derivatives

## B.1  Solution

The solution described here follows closely the method described in Mattingley and Boyd [2012], with small modifications for our LCP formulation above. The following is a small summary of that method, highlighting such differences.

In Equation 2, we formed the equivalent to the following system:

$$
\begin{aligned}
\mathcal{M}x + A^T y + G^T z + q &= 0 \\
Ax &= 0 \\
Gx + Fz + s &= m \\
s \geq 0, \ z \geq 0, \ s^T z &\geq 0.
\end{aligned}
\tag{24}
$$

To solve such system, after an initialization step (described in Mattingley and Boyd [2012]), we iteratively minimize the residuals from the equations above over the variables $x$, $s$, $z$ and $y$. At each iteration, if the stopping criteria (residual sizes and duality gap) are not met, we compute the affine scaling directions by solving the system

$$
\begin{bmatrix}
\mathcal{M} & 0 & G^T & A^T \\
0 & D(z) & D(s) & 0 \\
G & I & F & 0 \\
A & 0 & 0 & 0
\end{bmatrix}
\begin{bmatrix}
\Delta x^{\text{aff}} \\
\Delta s^{\text{aff}} \\
\Delta z^{\text{aff}} \\
\Delta y^{\text{aff}}
\end{bmatrix}
=
\begin{bmatrix}
-(\mathcal{M}x + A^T y + G^T z + q) \\
-(D(s)z) \\
-(Gx + Fz + s - m) \\
-(Ax)
\end{bmatrix}.
\tag{25}
$$

Then we compute the centering-plus-corrector directions

$$
\begin{bmatrix}
\mathcal{M} & 0 & G^T & A^T \\
0 & D(z) & D(s) & 0 \\
G & I & F & 0 \\
A & 0 & 0 & 0
\end{bmatrix}
\begin{bmatrix}
\Delta x^{\text{cc}} \\
\Delta s^{\text{cc}} \\
\Delta z^{\text{cc}} \\
\Delta y^{\text{cc}}
\end{bmatrix}
=
\begin{bmatrix}
0 \\
\sigma\mu\mathbf{1} - D(\Delta s^{\text{aff}})\Delta z^{\text{aff}} \\
0 \\
0
\end{bmatrix}.
\tag{26}
$$

where $\mu$ and $\sigma$ is defined in [Mattingley and Boyd, 2012]. We then update the variables by applying the following combined updates

$$
\begin{aligned}
x &:= x + \alpha(\Delta x^{\text{aff}} + \Delta x^{\text{cc}}) \\
s &:= s + \alpha(\Delta s^{\text{aff}} + \Delta s^{\text{cc}}) \\
z &:= z + \alpha(\Delta z^{\text{aff}} + \Delta z^{\text{cc}}) \\
y &:= y + \alpha(\Delta y^{\text{aff}} + \Delta y^{\text{cc}})
\end{aligned}
\tag{27}
$$

according to the step-size $\alpha$ defined in Mattingley and Boyd [2012].

## B.2  Derivatives

To obtain the derivatives, we use Equation 24 in a slightly modified form, such that at a solution point we have

$$
\begin{aligned}
\mathcal{M}x^\star + A^T y^\star + G^T z^\star + q &= 0 \\
Ax^\star &= 0 \\
D(z^\star)(Gx^\star + Fz^\star - m) &= 0.
\end{aligned}
$$

We use matrix differential calculus [Magnus and Neudecker, 1988] to take the differentials of these equations:

$$
\begin{aligned}
\mathrm{d}\mathcal{M}x^\star + \mathcal{M}\mathrm{d}x + \mathrm{d}A^T y^\star + A^T\mathrm{d}y + \mathrm{d}G^T z^\star + G^T\mathrm{d}z + \mathrm{d}q &= 0 \\
\mathrm{d}Ax^\star + A\mathrm{d}x &= 0 \\
D(Gx^\star + Fz^\star - m)\mathrm{d}z + D(z^\star)(\mathrm{d}Gx^\star + G\mathrm{d}x + \mathrm{d}Fz^\star + F\mathrm{d}z - \mathrm{d}m) &= 0,
\end{aligned}
\tag{28}
$$

which in matrix form is equivalent to

$$
\begin{bmatrix}
\mathcal{M} & G^T & A^T \\
D(z^\star)G & D(Gx^\star + Fz^\star - m) + F & 0 \\
A & 0 & 0
\end{bmatrix}
\begin{bmatrix}
\mathrm{d}x \\
\mathrm{d}z \\
\mathrm{d}y
\end{bmatrix}
=
\begin{bmatrix}
-\mathrm{d}\mathcal{M}x^\star - \mathrm{d}A^T y^\star - \mathrm{d}G^T z^\star - \mathrm{d}q \\
-D(z^\star)\mathrm{d}Gx^\star - D(z^\star)\mathrm{d}Fz^\star + D(z^\star)\mathrm{d}m \\
-\mathrm{d}Ax^\star
\end{bmatrix} .
$$

In this formulation, a given partial derivative, for example $\frac{\partial z^\star}{\partial q}$, can be found by substituting $\mathrm{d}q = I$, setting all other differential terms to zero, and solving for $\mathrm{d}z$. For the backpropagation algorithm, for a given backward pass vector with respect to the solution $x^\star$ to the LCP, say $\frac{\partial \ell}{\partial x^\star}$, we are interested in applying the chain rule to pass the derivatives further backwards, for example to find $\frac{\partial \ell}{\partial q}$ by multiplying $\frac{\partial \ell}{\partial x^\star}\frac{\partial x^\star}{\partial q}$. To simplify this process, let us first define the vector

$$
\begin{bmatrix}
d_x \\
d_z \\
d_y
\end{bmatrix}
:=
\begin{bmatrix}
\mathcal{M} & G^T & A^T \\
D(z^\star)G & D(Gx^\star + Fz^\star - m) + F & 0 \\
A & 0 & 0
\end{bmatrix}^{-T}
\begin{bmatrix}
\left(\frac{\partial \ell}{\partial x^\star}\right)^T \\
0 \\
0
\end{bmatrix} .
\tag{29}
$$

Then, we have that

$$
\frac{\partial \ell}{\partial x^\star}\mathrm{d}x =
\begin{bmatrix}
d_x \\
d_z \\
d_y
\end{bmatrix}^T
\begin{bmatrix}
-\mathrm{d}\mathcal{M}x^\star - \mathrm{d}A^T y^\star - \mathrm{d}G^T z^\star - \mathrm{d}q \\
-D(z^\star)\mathrm{d}Gx^\star - D(z^\star)\mathrm{d}Fz^\star + D(z^\star)\mathrm{d}m \\
-\mathrm{d}Ax^\star
\end{bmatrix} .
\tag{30}
$$

Now, by applying properties from matrix differential calculus we can propagate back the derivatives via chain rule to obtain the derivatives of our given backwards pass vector $\ell$ with respect to the inputs. For example, to obtain $\frac{\partial \ell}{\partial q} = \frac{\partial \ell}{\partial x^\star}\frac{\partial x^\star}{\partial q}$, we can see that

$$
\frac{\partial \ell}{\partial x^\star}\mathrm{d}x =
\begin{bmatrix}
d_x \\
d_z \\
d_y
\end{bmatrix}^T
\begin{bmatrix}
-\mathrm{d}q \\
0 \\
0
\end{bmatrix}
= -d_x^T \mathrm{d}q,
\tag{31}
$$

which implies $\frac{\partial \ell}{\partial x^\star}\frac{\partial x^\star}{\partial q} = -d_x$. The same procedure can be applied to the others quantities to arrive at the desired derivatives

$$
\begin{aligned}
\frac{\partial \ell}{\partial q} &= -d_x & \frac{\partial \ell}{\partial \mathcal{M}} &= -\frac{1}{2}(d_x x^T + x d_x^T) \\
\frac{\partial \ell}{\partial m} &= D(z^\star)d_z & \frac{\partial \ell}{\partial G} &= -D(z^\star)(d_z x^T + z d_x^T) \\
\frac{\partial \ell}{\partial A} &= -d_y x^T - y d_x^T & \frac{\partial \ell}{\partial F} &= -D(z^\star)d_z z^T .
\end{aligned}
\tag{32}
$$

## Footnotes

[2]Other examples available at `https://en.wikipedia.org/wiki/List_of_moments_of_inertia`

[3]We define the two dimensional cross product as the scalar $x \times y = x_1 y_2 - x_2 y_1$.