[Reviews · NeurIPS 2018]

Reviewer 1



The paper proposes to include an intermediate differentiable physics “engine” i.e. a novel parameterization of an intermediate layer in a neural network that respects the forward dynamics as governed by physics. The proposed method is closer to the popular “system identification” paradigm and involves learning the parameters of the engine via gradient-decent, optimizing the squared loss between observed and predicted frames. The paper shows results on some simple, interesting simulation domains and demonstrates the value of including such a module. Depending on the rebuttal for some of the questions below, I am happy to revise my ratings for this paper. Positives: - In general, the paper addresses an interesting and challenging question of incorporating more domain / structured knowledge into differentiable and learnable networks. - An interesting outcome of using a physics engine to transform the latent variable i.e. “step the physics engine” is that the dimensions of the latent space can now be assigned specific labels making the process interpretable to an extent. Comments and Questions: - L27 — such to to - L117 — v in v_{t+h} is not defined before. Although it is not hard to figure out, prefer including what it denotes at first use. - Given x, \phi_t, \phi_{t+h}., \hat{y} can be computed manually right ? What is the need to use a “decoder” network ? i.e. If I know ball is at position x_t and now has some velocity v for t seconds, I exactly know where the ball is at t+h and so, should be able to construct the next frame without having to “learn”. In this sense, I am not sure why one would burden the network with rendering the scene while also learning the physics of the process. - The L_{phys} reduces the promise of the differentiable physics engine. One would hope that by directly regressing on to the next frame (of course, one may argue that this is sparsely available however, simulation environments are being considered anyway) and parameterizing the intermediate layers as discussed, the model naturally (i.e. by only optimizing L_dec) learns a reasonable set of parameters that lead to fairly natural \phi vectors. In this sense, having to explicitly supervise these feels slightly disappointing. - The MLP baseline comparison is crucial to demonstrate the value of the method. From my understanding the provided results use labelled data (\phi_t, \phi_{t+h}) to train this MLP that sits in place of physics() module. Can the authors provide numbers for training such a module using only L_dec (and no labels) i.e. the dimensions of the latent variable are no longer “regularized” by the underlying physics ? Or in other words, if I am to predict \phi_{t+h} or \hat{y} directly, how well does the model perform ? In my opinion, this truly shows the gains due to not using the differentiable forward model. In the current setting, the MLP can be thought of as the physics engine only parametrized differently. - MSE can sometimes be inadequate for learning — for example, MSE between images can be driven down while the two images look very different to a human eye. Did the authors consider trying out other losses for comparing the frames ? After Rebuttal: With the additional evaluations and clarification of the discussions from the rebuttal into the paper, it will make for a reasonably interesting work to be accepted at NIPS. I am revising my rating accordingly.

Reviewer 2



The main idea of this submission follows in line with Amos et al.'s OptNet architecture, where a numerical (often iterative) routine is formulated in a differentiable manner with respect to its inputs. In this case, the optimization routine is an LCP solver, which is commonly used for solving for equations of motion, subject to friction and collision, in rigid body dynamics. Forward prediction using the differentiable LCP solver can be performed in a latent space, allowing rigid-body prediction from images. Strengths: - The paper is well-written, and I appreciated the literature overview of model-based RL and intuitive physics and differentiable optimization. kudos to the author's writing ability! - The use of an LCP solver to infer mass from interactions is clever. - I am surprised that the system works on a discrete-time system like Breakout Weaknesses: - Regarding novelty, I would like to see a comparison of the proposed method with simply using Mujoco's solver derivatives instead, which could conceivably be dropped into a neural network as well http://www.mujoco.org/book/programming.html#saDerivative and applied using the chain rule (some work required to wrap the C++ interface into a Pytorch layer, but I suspect it would yield a similar result). Questions: - To what extent can the rigid-body assumptions of LCP be relaxed, to afford better adaptation to model-free learning or real-world data (which does not conform well to LCP assumptions?) - My understanding is that rigid contact dynamics with nonlinear complementarity constraints are not invertible, and yet the LCP solver presented here has derivatives. Is there some relaxation going on here similar to the one proposed in Todorov '11 and Drumwright & Shell '09?

Reviewer 3



This paper proposed a way of implementing a differentiable physics engine based on Linear Complementarity Problem (LCP) where analytical gradients can be computed based on the primal-dual interior point method. As a result, it can be integrated as a module in an end-to-end learning pipeline, such as deep neural networks. Existing popular physical simulators, such as MuJoCo and Bullet, have also cast contact dynamics simulation as an optimization problem and resorted to techniques similar to LCP for solving the motion equations. The main (but limited) contribution of this work is to make analytical gradient computation more efficient with the interior point method. The problem of having a differentiable simulator is well motivated. It enables gradient-based learning/planning methods to backpropagate through dynamics. The authors illustrated the benefits of such a differentiable physical simulator in three types of experiments: 1) system identification, i.e., inferring the physical parameters from behaviors by backpropagating gradients to these parameters (e.g., mass); 2) using the physical simulator as a neural network layer for future prediction from simple visual observations; and 3) using it as a model to build controllers via iLQR. The proposed simulator adopted similar optimization techniques as in popular simulators, such as MuJoCo. Thus, it would suffer from the same issues caused by the “soft” contact model as in other LCP-based simulators, where contact is triggered when the distance of two contacting objects is below a threshold. Such approximation would lead to dynamics mismatch between the simulated contact physics and the real contact physics. However, the writing and analysis of this work can be significantly improved: A great amount of technical detail has been missing from the paper. Many of the symbols and notations in equations (e.q., Eq (1), (2), (3)) are undefined, and Sec. 3 deserved an expanded description of the LCP formulation as well as the primal-dual interior point method. For the sake of improving understanding, I would suggest the authors move some detailed motion equations to the supplementary material and instead provide more intuitive explanations of the method. In addition to the technical details, more analysis and comparison would be beneficial for understanding the speed and scalability of the proposed simulator. I would like to understand how fast the simulator can perform forward steps in some benchmarking tasks, compared to popular choices, e.g., MuJoCo, Bullet, and DART. Another question is the scalability of this simulator, such as how many contact points, objects, collision meshes the simulator can afford. The experiments have been conducted in tasks with a small number of articulated primitive shapes. The merit of such a physical simulator might be hindered by the limited size of problems it can simulate efficiently. Questions: Fig. 6: what is the main cause of the lower asymptotic performance of the model-based method in comparison to Double Q-Learning? Is it due to the approximate models built for the Cartpole and Breakout, or the iLQR solver used by the controller? In summary, this work introduces a way of implementing differentiable simulator with LCP techniques that allows efficient compute of analytical gradients. The technical novelty is a bit limited, but the experiment shows preliminary successes of such a differentiable physical simulator in several tasks. The major weakness of this paper is the lack of technical clarity and detailed analysis of this simulator, which has caused difficulty in judging the merit of this simulator in comparison to existing ones. As this paper can be significantly benefited from another round of revision, I am leaning towards rejecting this paper for now.